# Preliminary Research: Validation of the Method of Evaluating Resistance to Surface Wetting with Liquid of Protective Materials Intended for Polymer Protective Gloves

**DOI:** 10.3390/ijerph18179202

**Published:** 2021-08-31

**Authors:** Emilia Irzmańska, Aleksandra Jastrzębska, Magdalena Makowicz

**Affiliations:** 1Central Institute for Labour Protection—National Research Institute, Department of Personal Protective Equipment, 48 Wierzbowa, 90-133 Lodz, Poland; mamak@ciop.lodz.pl; 2Institute of Materials Science and Engineering, Lodz University of Technology, 1/15 Stefanowskiego, 90-924 Lodz, Poland; jastrzebska.aleksandra1@gmail.com

**Keywords:** polymers, wettability, protective gloves

## Abstract

The article presents validation argumentation of the novel method of evaluating resistance to surface wetting with different liquids of protective materials intended for polymer protective gloves based on the three parameters: water permeability index, non-wettability index and absorption index. Using our own method of evaluating resistance to surface wetting, it was shown that the knurled structure of the palm part of polymer protective gloves may inhibit transport of harmful and hazardous liquids outside the area of the protective glove. Currently, there is lack of objectifying methods for evaluation of surface wettability focused on the mentioned aspects. In view of the above facts, an original method for evaluating the resistance of protective materials to surface wetting with mineral oils and water has been invented and validated. It was assumed that the non-wettability index will be subjected to metrological analysis. Consequently, the validation process refers to this index. A precise assessment of the uncertainty budget of the individual components was obtained. On the basis of the obtained results, the measurement errors that may affect the quality and reliability of the test result performed in the laboratory were identified.

## 1. Introduction

Improvement of safety is the key aspect in the process of designing Personal Protective Equipment (PPE), including protective gloves [1]. The issue of safeness and durability of protective equipment is crucial not only during a global pandemic, but in the everyday life of many workers in the world [2,3,4]. Polymer gloves are one of the significant groups of gloves used in different working conditions.

Measuring the wettability of polymeric materials, designed for use in protective gloves, is the first step to evaluate their safety of use during work with wet and contaminated surfaces. Protective gloves resistant to chemical hazards designed to be used in such working conditions should have good adhesive and high wettability properties connected with superhydrophobicity. A surface with reversible adhesion properties leads to better grip and control during the manual work (holding or moving objects) and prevents the sudden slipping out of objects. However, hydrophobic properties of the glove material are connected with the possibility of transport of harmful and hazardous liquids off the glove in the workplace.

One of the possibilities leading to improving those properties is modification of polymer materials used to produce protective gloves. Discussed properties can be obtained by different chemical (chemical etching, chemical vapor deposition, wet chemical reaction, sol–gel processing) or physical (plasma treatment, laser microfabrication) modifications [5,6,7]. This field of study is focused on observing the mechanisms and structures existing in nature and implementing them into the technological inventions. Learning from sophisticated systems, morphologies and functions provide a wide range of ideas and concepts that are already copied by scientists and engineers from different fields of knowledge [8,9]. However, implementation of biological aspects is a very complex process that can be divided into three main steps: understanding the principles of the copying mechanism, implementation in the laboratory scale and the translation for selected application [10]. One of the most explored mechanisms of superhydrophobicity existing in nature is the “lotus-leaf effect”. A surface can be named as superhydrophobic when its contact angle (CA) is greater than 150° and the sliding angle is lower than 10° [11,12]. This phenomenon observed on the lotus leaf is strictly dependent on micro/nanostructures on its surface covered with hydrophobic wax [6]. In fact, describing this special property in 1997, Barthlott and Neinhuis initiated growth in research about superhydrophobic surfaces with extremely high contact angles (≥150°) [13,14]. The lotus effect is one of the many examples of superhydrophobic surfaces existing in nature. The other ones worth mentioning are petal roses, cicada wings, rice leaves and the water striders [11,12,15]. The list of potential application of such surfaces includes: self-cleaning, quick-drying, anti-fouling surfaces and coatings on different materials [16,17,18,19,20].

One of the biggest challenges in analyzing polymeric materials is studying the resistance to surface wetting with different liquids according to real working conditions. Wettability is controlled by the balance between the intermolecular interactions of the cohesive and adhesive type. Evaluation of the wettability of a particular surface in general can be affected by measuring the contact angle between the droplets of liquid with the surface [21]. This method is universally used for measuring the wettability of materials and products, including personal protective equipment [22,23,24]. Besides that, the method of the static water contact angle has some inconveniences and limitations. The measurement should be conducted very carefully with taking into consideration the following factors: contrast, lighting, focus and establishing the substrate baseline. Zimmermann et al. proved that even slightly different conditions and settings may affect the final result [25].

Another interesting, though less frequently used, method is imaging by using environmental scanning electron microscope (ESEM). This method can be used for observation of samples in water and analysis of wettability of the surface [26,27]. Besides of possibility of evaluation of the wettability of the surface at the micro/nano scale, this method has some inconveniences too. In the case of testing polymer materials, the specimens can be degraded because of the acceleration in the presence of water [28]. This is caused by high mobility of the radicals through the layer of water. This aspect limits the use of this method.

It is worth noting that a superhydrophobic surface means not only a high static angle, but also easiness for the rolling off of the liquid droplets. It is also important to take into consideration that wettability of material surface can be anisotropic; liquid droplets can easily roll along one direction but not along the other one [13]. Generally speaking, measuring the wettability of materials designed for protective gloves using only the static water contact angle method can be insufficient. Another aspect that has to be taken into consideration during measurement of the surface wettability is that different chemical substances can in different ways affect the polymeric materials. Resistance to surface wetting can be various depending on the character of the liquid [29]. In fact, the superoleophobic surface can be super-repellent to water because of the much higher surface tension of the water [30]. Modifications of the material can change the wetting character of different liquids, so the proper method designed for evaluation of this aspect is needed.

Currently, there is a standardized method of measuring the wettability of protective clothes. In the field of testing personal protective equipment, including polymeric protective gloves, it is necessary to expand the existing base of research on the wettability of such materials.

This novel method of measuring resistance to surface wetting of protective materials intended for polymer protective gloves is based on the method used for measuring resistance of materials to penetration by liquids according to the standard PN-EN ISO 6530:2008 [31]. In essence, it is a quick and simple method of evaluating resistance to surface wetting with different kind of liquids, e.g., water, oils and hazardous substances. The principle of this method is as following: a measured volume of a test liquid is applied in the form of a fine stream to the surface of material resting in an inclined gutter. The aim of the measurement is to determinate the respective proportions of the applied liquid that penetrate a test specimen and that are repelled by its surface. Obtained results indicate the potential of the material for use in the described field of application. Based on the series of measurements of polymeric materials, it was found that this method allows us to evaluate three parameters: the water permeability index (IP), non-wettability index (IR) and absorption index (IA) of tested materials. The results obtained from the series of measurements were compared and analyzed for correlations to determine whether this method is suitable for reliable evaluation of resistance to surface wetting with different liquids. The aim of this paper is the validation of the mentioned method.

## 2. Materials and Methods

### 2.1. Tested Materials

For the validation of the proposed method for surface wettability measurements, three types of polymers used commercially in the production of the protective gloves have been selected. The studied gloves were made of natural latex, nitrile rubber and butyl rubber. Gloves made of natural latex and butyl rubber have a rough surface of their palmar side. Their identification (sample name, producer and the material type) is listed in Table 1.

### 2.2. Surface Wettability Measurements Experimental Protocol

During this test, a minimum of three samples should be taken from the palm of the gloves. The next stage is to cut the 100 mm diameter discs of filter paper and foil and weigh both discs together with an accuracy of 0.01 g. The test polymer sample should also be weighed. On the gutter (see Figure 1), there should be placed successively: foil, filter paper and the sample. A very important note is the placing of the samples–when arranging the materials, their order should be the same as in the finished product. Subsequently, the cup should be mounted under the edge of the sample in order to collect the liquid flowing from the surface of the sample and a syringe should be attached, which is filled with the liquid used for testing. The test liquid (with a defined volume of 2.1 mL) is dropped on the surface of the sample, while the time is measured. After 3.0 s to 3.5 s (counted from the start of liquid flow), the person conducting the experiment should lightly rub the gutter to separate the drops hanging on the edge of the sample, remove the fixing tapes and remove the sample. Then, with the accuracy of 0.01 g, the filter paper with the foil under the sample, the dish with the liquid and the test sample need to be weighed. On this basis, it is possible to calculate the three parameters: water absorption index, non-wettability index and absorption index of the tested materials.

For the measurement, a laboratory balance Mettler AJ100 with the certificate no. 7W1.436.212819/2 was used (Mettler Wagen GmbH, Busigny, Switzerland). The syringe used for the test is manufactured by Polfast with a 10 mL volume.

The setup scheme for the measurements is provided in the Figure 1.

The measurement results are the wettability and non-wettability indexes, calculated for each test liquid, including the calculation of the water permeability, non-wettability and absorption indexes.


***Water permeability index (IP) [%]***


Permeability in this case has been considered as the process by which a chemical travels through pores, crevices or large openings in a fabric or final garment. The formula to calculate the IP is as follows:(1)$$I_{p}=\frac{M_{p}\cdot 100}{M_{t}}\;\left[\%\right]$$
where:

*M_p_*–mass of the liquid used for tests on the filter paper and on the foil [g];

*M_t_*–mass of the liquid used for the tests, applied to the sample [g].


***Non-wettability index (IR) [%]***


Non-wettability has been considered as the ability of a material to get rid of the liquid that is on its surface.
(2)$$I_{R}=\frac{M_{r}\cdot 100}{M_{t}}\;\left[\%\right]$$
where:

*M_r_*–mass of the liquid used for the tests, collected in the cup [g];

*M_t_*–mass of test liquid applied to the sample [g].


***Absorption index (IA) [%]***


Absorption has been described as the process of penetration of one substance (the liquid used in the test) into another substance forming a continuous phase (sample taken from the gloves).
(3)$$I_{A}=\frac{M_{A}\cdot 100}{M_{t}}\;\left[\%\right]$$
where:

*M_r_*–mass of the liquid used for the tests, absorbed by the tested material [g];

*M_t_*–mass of test liquid applied to the sample [g].

After the calculations, the values of the IP, IR and IA indicators should be rounded to one decimal place. The results of the measurements are the average values of the water absorption, non-wettability and absorption indexes.

## 3. Results

### 3.1. Validation and Statistical Analysis

The validation of the method for surface wettability measurements for protective gloves was based on the ISO standards (PN-EN ISO 6530:2008; PN-ISO 5725-1; PN-ISO 5725-2) [31,32,33], and it took into account parameters such as precision, measurement accuracy and uncertainty, as well as measurement errors. Comparison of the components of the standardized method according to EN ISO 6530:2008 [31] and the components of the validated method are provided in Table 2. It was conducted based on the series of results from laboratory test of the selected model polymeric gloves samples.

Estimation of the measurements uncertainty of the presented method involved the following components of uncertainty:

Standard deviation of repeatability and limit of repeatability, which was estimated using the formula:(4)$$S_{r}=\sqrt{\frac{\sum_{j=1}^{j}S_{j}^{2}}{j}}$$
where: *S_j_*—standard deviation of a single series of measurements, and *j*—number of measurement series. Moreover, the repeatability limits for all materials were calculated using the formula:(5)$$r=2.8\cdot S_{r}$$

Within-laboratory standard deviation of reproducibility was estimated using the formula
(6)$$S_{R}=\sqrt{\frac{\sum_{n=1}^{n}S_{n}^{2}}{n}}$$
where: *S_n_*—standard deviation of a single series of measurements, and *n*—number of measurement series. Moreover, the reproducibility limits for all the materials were calculated using the formula:(7)$$R=2.8\;\cdot \;S_{R}$$

Furthermore, measurements uncertainties were calculated, related to parameters such as the accuracy of measuring the diameter of the sample, the angle of the gutter, the accuracy of weighing the liquid used for the tests applied to the sample, the accuracy of weighing the liquid collected after the test and the dispersion of the measurement results.

Uncertainty related with accuracy of measurement of the sample diameter (*u*_1_).

The sample diameter was measured using a semi-rigid bar gauge, the expanded measurement uncertainty of which was ±0.3 mm at the confidence level of approx. 95% and the coverage factor was k = 2. The standard uncertainty *u*_1_ of the accuracy of the measurement of sample diameter was estimated using the formula:(8)$$u_{1}=\frac{0.3}{2}=0.15\;\left[mm\right]$$
where *w*_1_ (relative standard uncertainty) was $w_{1}=\frac{0.15}{100}=0.0015$.

Uncertainty related with the angle of the gutter (*u*_2_).

The gutter angle was measured with a universal protractor whose extended measurement uncertainty was ±3.2 at the confidence level of approx. 95% and the coverage factor was k = 2. The standard uncertainty *u*_2_ of the accuracy of the measurement of the angle of the gutter was estimated using the formula:(9)$$u_{2}=\frac{3.2}{2}=1.6\;\left[{}^{\circ}\right]$$

With $w_{2}=\frac{1.6}{2700}=0.000593$.

Uncertainty related with the accuracy of weighing the liquid applied to the sample and the liquid collected after test (*u*_3_).

The mass of the liquid was weighed on a non-automatic electronic analytical scale with an accuracy of ±0.01 g. The standard uncertainty *u*_3_ of the accuracy of weighing the liquid applied on the sample was estimated using the formula:(10)$$u_{3}=\frac{0.01}{\sqrt{3}}=0.00578\;\left[g\right]$$

With $w_{3}=\frac{0.00578}{2.73}=0.0021$.

The same procedure and formulas were used to calculate the standard uncertainty *u*_4_ of the accuracy of weighing liquid collected after test.

Uncertainty related with the dispersion of the measurements results:(11)$$u_{5}=\sqrt{S_{R}^{2}-\left(1-\frac{1}{n}\right)S_{r}^{2}}$$
where *n*—number of measurement repetitions, SR—standard deviations of within-laboratory reproducibility and *S_r_*—standard deviation of repeatability.

For the purpose of the detailed statistical analysis, the Cochran test was used to check and reveal uncertainties and outliers. Outliers were not taken into account for further calculations.

### 3.2. Results

The results of the measurements of the tested polymeric gloves resistance to surface wetting with liquid and the results of the non-wettability index are given in Table 3.

Based on the measurements conducted for the surface wettability of selected gloves, the measurement uncertainty for individual parameters and uncertainty budgets has been calculated and are presented in Figure 2, Figure 3 and Figure 4. They reflect the percentage contributions of the components of the described parameters to the compound measure of relative uncertainty.

Moreover, the relative uncertainty budgets for each studied material divided into components of measurement uncertainty for the measurements of the non-wettability of gloves are presented in Figure 5.

## 4. Discussion

Performing professional activities when hands are exposed to chemical (liquid) agents concerns mainly employees in the chemical, petrochemical, machinery, metal and automotive industries. One of the ways to ensure their safety is the use of appropriate protective gloves. Gloves should be made entirely of a hydrophobic polymer to effectively transport contaminants away from the glove area.

The hydrophobic properties of polymers for use in protective gloves are particularly important from the point of view of their safe use [34,35,36]. The hydrophobic nature of the surface reduces the risk of chemical contamination accumulating on the surface of the glove, which may not only affect the risk of harmful substances in the palm of the hand, but may also reduce the dexterity of movements performed in these gloves [37,38,39]. That is why it is extremely important to evaluate thoroughly the materials destined to be used for PPE in terms of their possible ways of interaction of their surfaces with substances that can be in contact with them [40,41,42].

There has been proposed a method of examination of surface wettability in terms of three parameters–water permeability index, non-wettability index and absorption index [31,43]. The correlation between them might serve as a useful analytical tool in terms of evaluation of the surface wetting of different materials and liquids, especially in simulated real-life conditions where different substances may affect studied materials in diverse ways.

In current laboratory practice, there is no such method of surface wettability examination that would take into consideration the evaluation of the mentioned parameters of surfaces contaminated with oils, chemicals, etc. [38]. Other authors conducted research in the field of the evaluation of dexterity tests for gloves protecting workers from chemicals and biological factors using the following tests: Minnesota-Turning Test, O’Connor Test Finger, the Purdue Pegboard test, the Crawford–Screws test and the O’Connor Tweezer test. The aim of this research was evaluating the twelve existing dexterity tests and identifying the ones with a high sensitivity degree. Research results prove that the Crawford–Screws test, Grooved Pegboard test, both Minnesota tests, the O’Connor Finger test, the test according to the standard ASTM F2010 and three Purdue tests are highly sensitive methods (56–67% sensitivity degree), in contrast to the standard method according to EN 420:2005:2003 (3% sensitivity degree) [44]. The research was carried out in the ergonomic aspect and found that the glove material has a decisive influence on the assessment of the effort put into grasping and pulling the cylinder in conditions of contaminated surface of protective glove.

It should be emphasized that there is no information in the literature describing an objective method of evaluating resistance to surface wetting with liquid for protective materials intended for polymer protective gloves. This aspect is crucial because of the fact that gloves without hydrophobic properties can decrease the safety of use, which may cause increasing numbers of accident rates at workplaces. Taking into account the above-mentioned aspects, authors conducted preliminary research on the development of a method of the influence of the type of glove material for work with chemically contaminated objects in the context of working conditions.

The developed procedure of the measurement of surface wetting allows us to determine the resistance of protective gloves—tight, made of rubber, plastics or knitted gloves coated with rubber or plastics—to surface wetting with different liquids. However, this method can also be used in the case of testing different materials every time where there is a need to evaluate the aspects like the water permeability, non-wettability index and absorption index of materials.

Validation of the method of evaluating resistance to surface wetting with liquid of the protective materials is not directly described in the literature. There are a significant number of papers that concern the indirect evaluation of polymer materials without indicating the direction of their application [45,46].

Although the concept of working comfort in gloves is not measurable precisely with the available devices while performing professional activities, there are methods of classifying these features, which, from the glove user’s point of view, may be a quantifier of usability and safety. It is important with gloves for special purposes, e.g., protective gloves, intended for special tasks in particularly chemically difficult working conditions. Achieving a compromise between the safety of use and the intended use of the gloves requires the optimization of the parameters determining the metrological degree of their usefulness. There are close relationships between the construction of the glove, the material from which it is made and physical factors that significantly affect the hydrophobicity of the material. These factors make it possible to match a given type of glove to professional activities. During the analysis of the wetting properties of the surface of polymer gloves, the empirical values related to water permeability, non-wettability index and absorption index of materials was taken into consideration. The uncertainty budgets indicate that accuracy of the weight of the liquid used during the measurement has the highest influence on the scatter of the measurements results.

## 5. Conclusions

The conducted validation of the developed, novel measurement method has shown that this method is well-targeted, and the analysis of the revealed uncertainties and error estimation allows for conclusions about its suitability for solving the research problem, which is the assessment of work safety comfort in polymer protective gloves from the user’s point of view. The measurements conducted with high precision and the repeatability of control for the non-wettability index may allow for the appropriate selection of protective gloves in the foreseeable future, taking into account the aspects of the types of contamination with chemical hazards such as oils. The validation of the method in terms of metrology confirmed that the obtained uncertainty budgets allow for an effective and, what is more, objectivized assessment of the comfort of using polymer protective gloves in the workplace with the chemical hazard.

## Figures and Tables

**Figure 1 ijerph-18-09202-f001:**
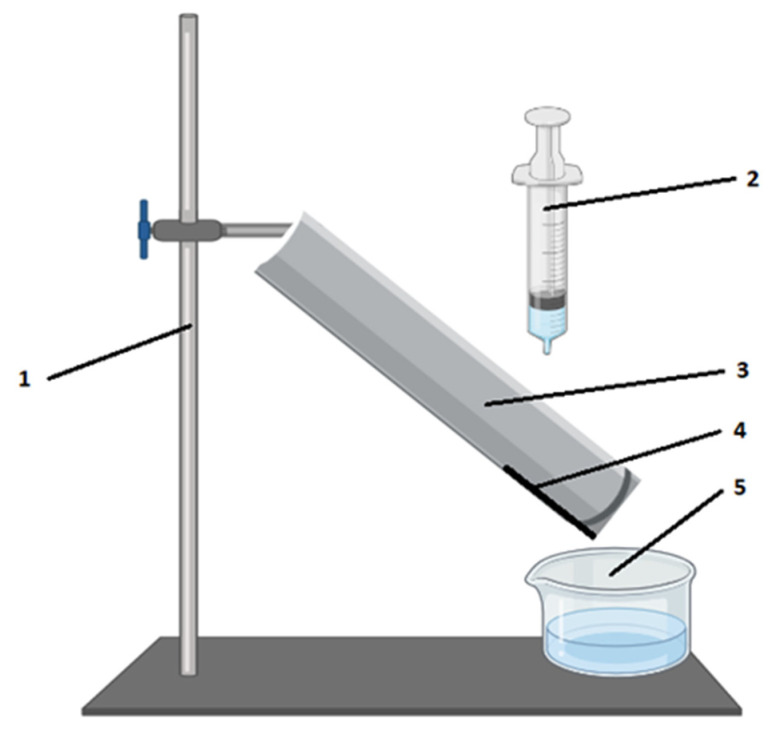
Setup for wettability measurements. 1—metal base for fixing the gutter at an angle of inclination (45°); 2—a syringe with a volume of (10 ± 0.5) mL with a needle with an inside diameter (0.8 ± 0.02) mm; 3—rigid semi-cylindrical PVC gutter with an internal diameter (70 ± 5) mm and length (190 ± 5) mm; 4—tested sample; 5—a beaker with measurement liquid. [Figure prepared by using Professional version of BioRender Program].

**Figure 2 ijerph-18-09202-f002:**
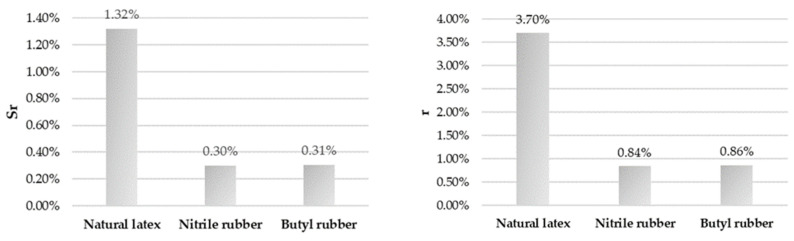
Uncertainty budget for the repeatability of the performed measurements (on the **left**) and the repeatability limits (on the **right**).

**Figure 3 ijerph-18-09202-f003:**
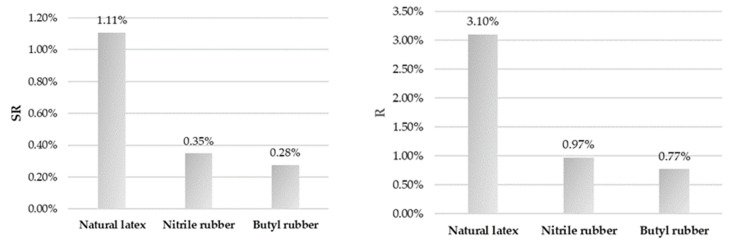
Uncertainty budget for the reproducibility of the performed measurements (on the **left**) and the reproducibility limits (on the **right**).

**Figure 4 ijerph-18-09202-f004:**
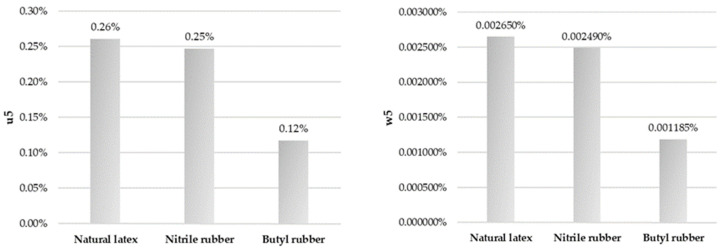
Uncertainty budget for the dispersion of the performed measurements (on the **left**) and the relative dispersion limits (on the **right**).

**Figure 5 ijerph-18-09202-f005:**
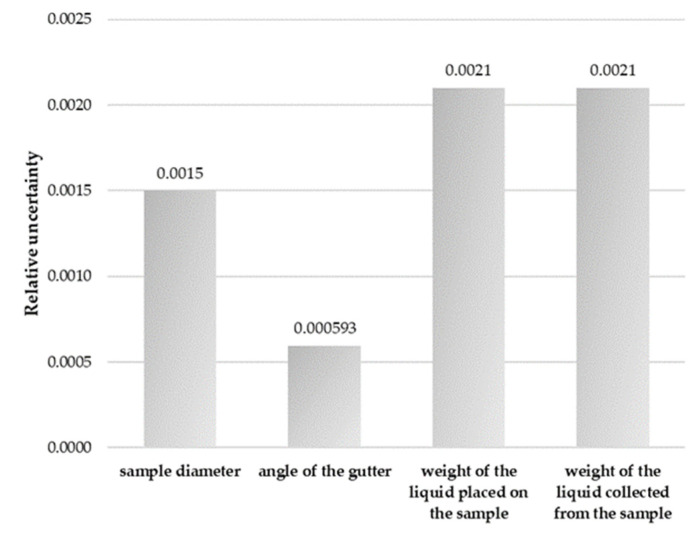
Relative uncertainty of the components of the performed measurements.

**Table 1 ijerph-18-09202-t001:** Gloves used in the study.

Type of Material	Sample Designation	Photographs	Thickness [mm]	Contact Angle for Water [°]	Work of Adhesion [mJ/m^2^]
Palmar Surface	Dorsal Surface	Palmar	Dorsal
**Natural Latex**	G1	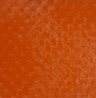	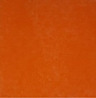	0.92 ± 0.01	64.97 ± 8.61	67.47 ± 5.19	118.57
**Nitrile Rubber**	G2	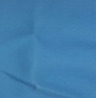	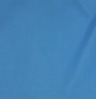	0.20 ± 0.01	32.48 ± 0.98	25.46 ± 12.14	152.32
**Butyl Rubber**	G3	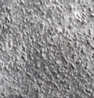	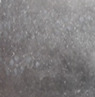	0.67 ± 0.02	79.21 ± 9.68	76.49 ± 3.77	153.36

**Table 2 ijerph-18-09202-t002:** Comparison of the components of validated method and method according to EN ISO 6530:2008 [31].

Component	Validated Method	EN ISO 6530:2008
**Gutter**	rigid PCV gutter, of semi-cylindrical shape, with an internal diameter (70 ± 5) mm, length (190 ± 2) mm and inclination 45°	rigid transparent gutter, of semi-cylindrical shape, with an internal diameter (125 ± 5) mm, length (300 ± 2) mm and inclination 45°
**Needle**	needle (0.8 ± 0.02) mm	hypodermic needle, bore (0.8 ± 0.02) mm
**Syringe**	syringe with volume (10 ± 0.5) mL	syringe or other leak-free attachment to the needle, capable of delivering (10 ± 0.5) cm^3^ of test liquid
**Beaker**	small beaker	
**Foil**	foil resistant to the test liquid	transparent film resistant to the test liquid
**Paper**	absorbent paper, 0.15 mm to 0.3 mm thick	
**Stopwatch**	stopwatch, accurate to 0.1 s	
**Balance**	balance, accurate to 0.01 g	
**Test specimens**	three specimens from palmar surface of protective gloves with diameter 100 mm	six specimens of (360 ± 2) mm by (235 ± 5) mm from the clothing or sample of material

**Table 3 ijerph-18-09202-t003:** Measurements of the non-wettability index of the tested specimens.

Sample	Part	Measurement Series j	Results [%]			Average	Standard Deviation S_d_	Variance S^2^_rj_
			n = 1	n = 2	n = 3			
**Latex rubber**	**Palmar**	**Series 1**	98.75	98.46	96.75	97.99	1.081	1.169
		**Series 2 (after 1 month)**	99.50	97.00	96.83	97.78	1.495	2.235
		**Series 3 (after 3 months)**	97.32	99.00	97.25	97.86	0.991	0.982
	**Dorsal**	**Series 1**	96.34	99.50	98.75	98.20	1.651	2.756
		**Series 2 (after 1 month)**	97.32	99.75	99.25	98.77	1.283	1.646
		**Series 3 (after 3 months)**	96.83	99.25	98.78	98.29	1.283	1.646
	**St.dev. of the within-laboratory reproducibility of the S_Rn_ series [%]**		1.203	0.999	1.114			
	**Intra-laboratory reproducibility variance S^2^_Rn_ [%^2^]**		1.447	0.998	1.241			
**Nitrile rubber**	**Palmar**	**Series 1**	98.50	99.25	99.50	99.08	0.520	0.271
		**Series 2 (after 1 month)**	99.00	99.00	99.50	99.17	0.289	0.083
		**Series 3 (after 3 months)**	99.25	99.00	99.25	99.17	0.144	0.021
	**Dorsal**	**Series 1**	99.50	99.75	99.75	99.67	0.144	0.021
		**Series 2 (after 1 month)**	97.56 *	99.75	99.75	99.75	0	0
		**Series 3 (after 3 months)**	99.25	100.00	99.50	99.58	0.382	0.146
	**St.dev. of the within-laboratory reproducibility of the S_Rn_ series [%]**		0.379	0.431	0.188			
	**Intra-laboratory reproducibility variance S^2^_Rn_ [%^2^]**		0.144	0.185	0.035			
**Butyl rubber**	**Palmar**	**Series 1**	98.75	99.50	98.75	99.00	0.433	0.188
		**Series 2 (after 1 month)**	98.75	99.00	99.25	99.00	0.250	0.063
		**Series 3 (after 3 months)**	99.00	99.00	99.00	99.00	0	0
	**Dorsal**	**Series 1**	98.75	99.50	99.00	99.08	0.382	0.142
		**Series 2 (after 1 month)**	99.25	99.49	99.50	99.41	0.141	0.020
		**Series 3 (after 3 months)**	98.75	99.25	99.50	99.17	0.382	0.146
	**St.dev. of the within-laboratory reproducibility of the S_Rn_ series [%]**		0.210	0.244	0.354			
	**Intra-laboratory reproducibility variance S^2^_Rn_ [%^2^]**		0.044	0.060	0.125			

* The marked value was considered as an outlier and was excluded in further calculations.

## Data Availability

Not applicable.

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
