# Peer review of "Preliminary Research: Validation of the Method of Evaluating Resistance to Surface Wetting with Liquid of Protective Materials Intended for Polymer Protective Gloves"

_ijerph, 2021, doi:10.3390/ijerph18179202_

Round 1

Reviewer 1 Report

Page 1, lines: 34-35: Protective gloves resistant to chemical hazards designed to be used in such working conditions should have good adhesive and high wettability properties connected with high contact angle (≥150°) - Reference and explanation needed.

Page 2, lines 93-94: Currently, there is lack of standardized method of measuring the wettability of polymeric materials that take into consideration mentioned features. - This is is an overhasty statement. 

Table 1. The selection of materials/structures for the study is poor. It would be interesting to see how work of adhesion and contact angle change depending on the roughness of the surface. Thus, on one hand surface profiling is needed, and, on the other hand, a number of tested surfaces should be larger (e.g., three different butyl rubber surfaces), etc. 

Can you please, discuss Figures 2-5? Comments are missing, the reader does not know the meaning of the results and similarity & dissimilarity between tested groups.

Author Response

Dear Reviewer,

Reviewer 2 Report

  1. The authors should present in the article an analysis and comparison of their own method with the methods presented in the standards: «ISO 17771:2003 Plastics — Thermoset moulding compounds — Determination of the degree of fibre wetting in SMC», «ISO 8296:2003 Plastics — Film and sheeting — Determination of wetting tension», «ISO 14419:2010 Textiles - Oil repellency – Hydrocarbon resistance test».
  2. What is the uniqueness of the task that the authors solve? It should be taken into account that ISO-methods have a large volume of evidence-based tests for various materials. Line 93-94: The authors should formulate more convincingly the novelty that the purpose of the presented research has.
  3. The authors should provide an additional table with a comparison of the components of the EN ISO 6530:2008 method and the components of the new method proposed by the authors.
  4. The authors reported a large value of the surface roughness parameters for wetting [8-9]. The authors should provide additional technical data on the surface roughness parameters of the test samples in table 1. It is necessary to present the chemical composition of the materials of the test samples.
  5. Figure 1 shows an element in the form of a measuring volume for a liquid, which is called "a weighting plate". It is necessary to bring the drawing and the physical meaning into line.
  6. The authors should provide in the article an additional table with technical parameters and data on the accuracy of measurements for all components of the measuring system, including the parameters of measuring instruments and auxiliary materials (paper, foil, etc.).
  7. The authors presented an important criterion for protective gloves (line 34-36): « Protective gloves resistant to chemical hazards designed to be used in such working conditions should have good adhesive and high wettability properties connected with high contact angle (≥150°)». In the article, it is necessary to explain how the presented method takes into account an important wetting angle of more than ≥150° (Figure 1 shows the research scheme for wetting at 45°).
  8. In the article, it is necessary to present the parameters of aggressive liquids that were used to validate the method (except water).

Author Response

Dear Reviewer,

Round 2

Reviewer 1 Report

No further comments. 

Reviewer 2 Report

The article can be recommended for publication.